# Comparing in-person, blended and virtual training interventions; a real-world evaluation of HIV capacity building programs in 16 countries in sub-Saharan Africa

E. Kiguli-Malwadde[1], M. Forster[2], A. Eliaz[3], J. Celentano[2], E. Chilembe[4], I. D. Couper[5], E. T. Dassah[6], M. R. De Villiers[5], O. Gachuno[7], C. Haruzivishe[8], J. Khanyola[9], S. Martin[2], K. Motlhatlhedi[10], R. Mubuuke[11], K. A. Mteta[12], P. Moabi[13], A. Rodrigues[14], D. Sears[15], F. Semitala[10], D. von Zinkernagel[2], M. J. A. Reid[2,15]*, F. Suleman[16]

1 African Center for Global Health and Social Transformation, Kampala, Uganda, 2 Institute for Global Health Sciences, University of California, San Francisco, California, United States of America, 3 Department of Medicine, Stanford University School of Medicine, Stanford, California, United States of America, 4 Kamuzu College of Nursing, University of Malawi, Kamuzu, Malawi, 5 Department of Global Health, Ukwanda Centre for Rural Health, Stellenbosch University, Stellenbosch, South Africa, 6 School of Public Health, Kwame Nkrumah University of Science and Technology, Kumasi, Ghana, 7 Faculty of Medicine, Department of Obstetrics and Gynecology, University of Nairobi, Nairobi, Kenya, 8 Faculty of Health Sciences, University of Zimbabwe College of Health Sciences, Harare, Zimbabwe, 9 University of Global Health Equity, Kigali, Rwanda, 10 Faculty of Medicine, Department of Family Medicine and Public Health, University of Botswana, Botswana, 11 School of Medicine, Makerere University, Kampala, Uganda, 12 Kilimanjaro Christian Medical University College, Kilimanjaro, Tanzania, 13 Scott College of Nursing, Morija, Lesotho, 14 Faculty of Medicine, Eduardo Mondlane University, Maputo, Mozambique, 15 Department of Medicine, Division of Infectious Diseases, University of California, San Francisco, California, United States of America, 16 School of Health Sciences, University of KwaZulu-Natal, Durban, South Africa

☯ These authors contributed equally to this work.
* michael.reid@ucsf.edu

**Data Availability Statement:** Data are available in a public, open access repository. Extra data can be

## Abstract

We sought to evaluate the impact of transitioning a multi-country HIV training program from in-person to online by comparing digital training approaches implemented during the pandemic with in-person approaches employed before COVID-19. We evaluated mean changes in pre-and post-course knowledge scores and self-reported confidence scores for learners who participated in (1) in-person workshops (between October 2019 and March 2020), (2) entirely asynchronous, Virtual Workshops [VW] (between May 2021 and January 2022), and (3) a blended Online Course [OC] (between May 2021 and January 2022) across 16 SSA countries. Learning objectives and evaluation tools were the same for all three groups. Across 16 SSA countries, 3023 participants enrolled in the in-person course, 2193 learners participated in the virtual workshop, and 527 in the online course. The proportions of women who participated in the VW and OC were greater than the proportion who participated in the in-person course (60.1% and 63.6%, p<0.001). Nursing and midwives constituted the largest learner group overall (1145 [37.9%] vs. 949 [43.3%] vs. 107 [20.5%]). Across all domains of HIV knowledge and self-perceived confidence, there was a mean increase between pre- and post-course assessments, regardless of how training was

accessed via the Dryad data repository at http://datadryad.org/ with the doi:10.7272/Q6WQ021N.

**Funding:** EKM, MF, JC, EC, IDC, ETD, MRD, OG, CH, JK, SM, LM, AKM, PM, AR, DS, FS, DVZ, MJAR and FS were all supported by the US Health Resource Services Administration, through a PEPFAR grant. The funder had no role in the study and was not involved in the study design, collection or analysis of data, nor in the decision to publish or prepare this manuscript.

**Competing interests:** Judy Khanyola is an author and also a editor for PLOS GPH. This does not alter our adherence to PLOS ONE policies on sharing data and materials.

delivered. The greatest percent increase in knowledge scores was among those participating in the in-person course compared to VW or OC formats (13.6% increase vs. 6.0% and 7.6%, p<0.001). Gains in self-reported confidence were greater among learners who participated in the in-person course compared to VW or OC formats, regardless of training level (p<0.001) or professional cadre (p<0.001). In this multi-country capacity HIV training program, in-person, online synchronous, and blended synchronous/asynchronous strategies were effective means of training learners from diverse clinical settings. Online learning approaches facilitated participation from more women and more diverse cadres. However, gains in knowledge and clinical confidence were greater among those participating in in-person learning programs.

## Introduction

Even before the COVID-19 pandemic, there was a pressing need to expand the health workforce and optimize team-based care across sub-Saharan Africa (SSA), especially in settings of scarce health resources.] While causing unprecedented disruptions to clinical care and health professions training opportunities, the COVID-19 pandemic presented a unique opportunity to rapidly scale new training modalities, including maximizing the use of online learning platforms [1–6]. Numerous African and international training programs developed and implemented remote educational programs to address safety concerns regarding COVID-19 transmission and to ensure continuity of training programs in spite of government interventions to reduce infection and transmission [3,7–10]. Moreover, many higher education institutions in Africa adopted emergency remote teaching and web-based learning using blended-learning strategies to reduce the physical presence of students on campus [4,11]. One such example was the Strengthening Inter-professional Education in HIV (STRIPE HIV) initiative, which rapidly transitioned its in-person training program to online [12]. STRIPE HIV is an HIV training program funded by the United States government through the President's Emergency Plan for AIDS Relief (PEPFAR), that includes a diverse network of health professional training institutions across SSA, led by the African Forum for Research and Education in Health (AFREhealth) and the University of California, San Francisco [13]. At the start of the COVID-19 pandemic, the STRIPE team rapidly implemented remote educational programming, including developing a learning management system (LMS) accessible to health professions at affiliated training institutions across SSA [1]. The internet-based educational programs consisted of inter-professional HIV training content for pre-service and in-service learners from diverse professional cadres; while the format was altered to accommodate online learning, the curriculum was the same as that employed pre-COVID [12,13].

In advance of implementing this online approach to training, limited research comparing the effectiveness of in-person and remote educational formats across SSA had been undertaken [6,14]. Previous research has evaluated the impact of numerous capacity-building interventions to target individual healthcare provider behavior in low and middle-income countries (LMICs) [14–17], but very little research has evaluated the comparative benefits of online capacity-building interventions in such settings. The in-person and online remote educational programs developed by STRIPE HIV and AFREhealth provide a unique opportunity to evaluate whether transitioning to online learning significantly impacted inter-professional training in HIV care. As such, we sought to compare the impact of the in-person and online formats of inter-professional education on knowledge and confidence among learners of diverse cadres across fourteen countries in SSA.

## Methods

### Curriculum design and implementation: Year one in-person program

A case-based curriculum in inter-professional HIV training was developed and implemented during year one (Y1) of STRIPE HIV's educational program, previously described.[12] The program was developed using Kern's six-step approach by a collaborative team of local and international HIV practitioners and health professions education experts [18]. The inter-professional HIV training included 17 case-based modules with at least four required modules delivered in person over a two-day workshop. The program was initially implemented across 14 countries in SSA between October 2019 and April 2020.

### Curriculum design and implementation: Year two online synchronous program

In year two (Y2), which coincided with the spread of COVID-19 across SSA, participating academic institutions opted for one of two learning modalities based on Internet access and existing expertise using educational technologies. Many schools opted to pursue employ a Virtual Workshop (VW) format, which consisted of synchronous online workshops. This approach utilized a revised version of the in-person training content from Y1, adapted for delivery over Zoom and taught by both local and international facilitators, but incorporating both small-group activities and large-group discussions like the in-person approach. Learners from fourteen countries, including learners from several countries that had not previously (Algeria, Madagascar, Mozambique, and Somalia), participated in the Virtual Workshop beginning in May 2021.

### Curriculum design and implementation: Year two online blended program

Several other academic institutions opted to teach the same modular content using an online approach that included blended asynchronous and synchronous content (hereafter referred to as the Online Course [OC]). The asynchronous content in this blended program was completed independently online using a Moodle-based learning management system. After completion of the asynchronous content, learners took part in synchronous inter-professional sessions carried out on Zoom. Learners participated in a synchronous session associated with each of the modules using a flipped classroom approach [19]. The Online Course was implemented across six countries in SSA beginning in June 2021.

As outlined in our previous work [12,13], each STRIPE collaborating institution recruited learners from different cadres (e.g. nursing, medicine, pharmacy, etc.), stages of professional development (pre-service, early graduate, etc.) and clinical settings (clinic, hospital, classroom, etc.) according to local training priorities. Moreover, collaborating institutions employed different, non-standardized recruitment strategies to optimize the participation of learners, with some institutions providing airtime or access to the Internet to learners who participated in VW or OC courses.

### Study design & subjects

We compared learner knowledge and confidence across the three educational programs: the In-Person Training (Y1), the Virtual Workshop (Y2), and the Online Course (Y2). Subjects were learners who completed the STRIPE registration survey, as well as pre-test and post-test knowledge and confidence assessments. We evaluated self-reported learner demographics including gender identity, training level, health profession, and country of residence. Pre-service learners were defined as learners enrolled in a health profession training institution and working towards their degree. In-service or post-graduate, clinical providers were learners

who had already graduated from their health professions training program and engaged in clinical practice. Additionally, we defined early in-service learners as learners who had graduated from training fewer than 12 months prior to participating in the STRIPE HIV course.

To compare the three learning approaches, we evaluated data from four training modules that utilized the same pre-test and post-test assessments across the three educational formats. The four case-based modules were titled *New HIV Diagnosis and Antiretroviral Therapy (ART) Initiation in a Woman of Childbearing Age*, *Management of HIV-TB Co-Infection*, *PMTCT & Care for the Pregnant Woman with HIV*, and *Care for the Paediatric Patient with HIV*. The specific knowledge and confidence assessment questions from each module utilized in the analysis are shown in S1 Table. Since the learning objectives, key content, and learning activities were the same or very similar, regardless of which approach was employed, the learning time for these modules was similar for the In-Person, OC and VW formats.

## Knowledge assessment

We evaluated learning among participants by assessing the mean difference in pre-test and post-test knowledge scores and percent change in knowledge scores. For Y1, data were collected during the period of training from October 2019 to March 2020. For Y2, data were collected from May 2021 to January 2022. We compared mean knowledge differences and percent increase across the educational programs and stratified knowledge differences by health profession and training level.

## Confidence assessment

We evaluated confidence among participants by assessing the mean difference in pre-test and post-test confidence scores in three arenas: clinical confidence, confidence in working as part of an inter-professional team (IP Confidence), and confidence in implementing quality improvement strategies (QI Confidence). Across these three confidence domains, participants were invited to complete a series of questions asked on a Likert scale of 1 ("I feel uncomfortable with this topic/need supervision from my supervisor") to 4 ("I feel very comfortable with this topic/without supervision as though in independent practice"). Fifteen questions assessed clinical confidence, two questions assessed IP confidence and one question addressed confidence to leverage QI tools in clinical practice. These confidence scores were summed for each participant, and the mean differences in confidence scores were compared across the three educational programs, stratified by health profession and training level, using Analysis of Variance (ANOVA) and Tukey's Honest Significance Tests as outlined below.

## Statistical analysis

We compared learner demographics using Pearson chi-squared tests. Wilcoxon signed-rank test was used to identify baseline pairwise differences between the pre-test and post-test means for each educational program overall, as well as by training level and health profession. We used Analysis of Variance (ANOVA) and Tukey's Honest Significance Test (HSD) to compare mean differences in knowledge and confidence between the three educational programs. Analyses were carried out using RStudio (2022.07.1+554). Additionally, $p < 0.05$ was considered statistically significant.

## Ethics statement

The design of the training program, including the topics covered and the format of the training, was informed by input from focus-group discussions with patient groups, learners (both

pre-service and early career professionals) and HIV educators from a variety of settings in SSA, and has been previously described [13,20]. Assessment tools, evaluating learners knowledge and confidence, were also piloted with a subset of multidisciplinary learners before the full program was launched. All learners were given access to their pre and post score test results, via the program's website. In addition, aggregate, site-level evaluation data were also posted on the program's website. The protocol for this project was reviewed and approved by the University of California, San Francisco's Institutional Review Board (IRB) in San Francisco, California. Verbal consent was required at the time of participation in the study as approved by the IRB (protocol #: 19–28,447).

## Findings

### Learner demographics

During Y1, from October 2019 to March 2020, 5027 participants from 12 countries (Botswana, Ethiopia, Ghana, Kenya, Lesotho, Malawi, Nigeria, South Africa, Tanzania, Uganda, Zambia, and Zimbabwe) participated in the In-Person Course, and 3023 of those learners completed pre-test and post-test assessments. During Y2, from June 2021 to November 2022, 7107 learners from 15 countries in sub-Saharan Africa (Algeria, Botswana, Ethiopia, Ghana, Kenya, Lesotho, Madagascar, Malawi, Mozambique, Nigeria, South Africa, Somalia, Tanzania, Zambia, and Zimbabwe) enrolled in the VW and OC programs and 3477 of those learners completed the pre-test and post-test assessments. Among the 3477 remote learners included in the study, 2595 completed the VW training and 628 completed the OC training (Table 1). Complete demographic data was available for 3022, 2193 and 527 learners in the In-person, VW and OC respectively, while pre- and post-course evaluative data was available for 3027, 2595 and 629 participants in each of these groups.

### Gender

Among Y1 participants, 1570 (51.9%) identified as female, 1281 (42.4%) identified as male, and 172 (5.7%) identified as other. The proportions of female learners in the Y2 Virtual Workshop and Y2 Online Course were significantly greater than the Y1 in-person program (60.1% [n = 1319] and 63.6% [n = 335] vs. 51.9% [n = 1570], p<0.001).

### Training level

Among learners in the Y1, 1755 (58.1%) identified as pre-service, 727 (24.1%) identified as new providers, less than 12 months from graduation, and 540 (17.9%) identified as post-graduate providers, more than 12 months from graduating their training program (Table 1). The proportions of in-service new providers in the Y2 Virtual Workshop and Y2 Online Course were significantly lower than the Y1 in-person program (13.3% [n = 291] and 12.6% [n = 66] vs 24.1% [n = 727], p<0.001).

### Health professions

In both the Y1 In-person program and remote Y2 educational programs, at least four different health professions participated in the training. Health professions included laboratory, medical, nursing/midwifery, pharmacy, and other, (We included physical therapists and nutritionists in the "other" category given the small number of learners from those professional cadres.) Nursing/midwifery learners constituted the largest proportion of learners in the In-person (n = 1145 [37.9%]) and VW (n = 949 [43.3%]) courses, while medical providers constituted the largest group in the OC (n = 140 [26.8%]).

**Table 1. Demographic summary of Y1 In-person, Y2 Virtual Workshop, and Y2 Online Course learners included in the study.**

|  | Y1 In-Person | | Y2 Virtual Workshop | | Y2 Online Course | |
|---|---|---|---|---|---|---|
|  | No. | (%) | No. | (%) | No. | (%) |
| **Gender Identity** | 3023 | (100.0) | 2193 | (100.0) | 527 | (100.0) |
| Female | 1570 | (51.9) | 1319 | (60.1) | 335 | (63.6) |
| Male | 1281 | (42.4) | 873 | (39.8) | 192 | (36.4) |
| Additional | 172 | (5.7) | 1 | (0) | 0 | (0) |
| **Current Training Level** | 3022 | (100.0) | 2193 | (100.0) | 523 | (100.0) |
| Pre-service learner | 1755 | (58.1) | 1518 | (69.2) | 299 | (57.1) |
| Post-graduate new provider* | 727 | (24.1) | 291 | (13.3) | 66 | (12.6) |
| Post-graduate provider** | 540 | (17.9) | 384 | (17.5) | 158 | (30.2) |
| **Current Health Profession** | 3023 | (100.0) | 2193 | (100.0) | 523 | (100.0) |
| Laboratory | 365 | (12.1) | 86 | (3.9) | 117 | (22.4) |
| Medical | 902 | (29.8) | 582 | (26.5) | 140 | (26.8) |
| Nursing/midwifery | 1145 | (37.9) | 949 | (43.3) | 107 | (20.5) |
| Pharmacy | 312 | (10.3) | 373 | (17.0) | 95 | (18.2) |
| Other | 299 | (9.9) | 203 | (9.3) | 64 | (12.2) |
| **Country** | 3022 | (100.0) | 2595 | (100.0) | 628 | (100.0) |
| Algeria | 0 | (0) | 14 | (0.5) | 3 | (0.5) |
| Botswana | 174 | (5.8) | 2 | (0.1) | 113 | (18.0) |
| Ethiopia | 50 | (1.7) | 54 | (2.1) | 0 | (0) |
| Ghana | 733 | (24.3) | 542 | (20.9) | 0 | (0) |
| Kenya | 1 | (0.0) | 326 | (12.6) | 1 | (0.2) |
| Lesotho | 130 | (4.3) | 686 | (26.4) | 0 | (0) |
| Madagascar | 0 | (0) | 1 | (0) | 0 | (0) |
| Malawi | 512 | (16.9) | 254 | (9.8) | 73 | (11.6) |
| Mozambique | 0 | (0) | 17 | (0.7) | 0 | (0) |
| Nigeria | 192 | (6.4) | 0 | (0) | 252 | (40.1) |
| Somalia | 0 | (0) | 1 | (0) | 0 | (0) |
| South Africa | 323 | (10.7) | 128 | (4.9) | 1 | (0.2) |
| Tanzania | 50 | (1.7) | 194 | (7.5) | 0 | (0) |
| Uganda | 635 | (21.0) | 0 | (0) | 0 | (0) |
| Zambia | 110 | (3.6) | 133 | (5.1) | 0 | (0) |
| Zimbabwe | 112 | (3.7) | 243 | (9.4) | 186 | (35.6) |

*Y1*: Year one; *Y2*: Year two. *New post-graduate provider: Provider participating in training less than 12 months since graduating from health profession training program; **post-graduate provider: Provider participating in training at least 12 months or more since graduating from health profession training program.

## Knowledge assessment

Across all subgroups of learners, HIV knowledge scores increased between the pre- and post-course tests; the mean difference in pre vs. post-test knowledge scores among participants in Y1 was 1.2 compared to 0.5 among VW participants and 0.7 among OC participants (S3 Table). Moreover, the percent increase in knowledge scores was significantly lower for those who participated in either the VW or OC formats than among learners who participated in the Y1 In-person training (+6.0% and +7.6% vs +13.6%, p<0.001) (Table 2).

Across the three training levels, the increase in knowledge scores was greatest for participants in the Y1 In-person course compared to either VW or OC across stages of professional development (Y1 in-person pre-service learners experienced a 14.1% [+1.3 points] mean increase in knowledge vs. 5.1% [+0.5 points] for Y2 VW learners vs. 7.4% [+0.6 points] for OC

**Table 2. Total learner mean knowledge score differences during Y1 in-person Training, Y2 Virtual Workshop, and Y2 Online Course by Total Score, Training Level, and Health Profession compared using ANOVA.**

| | Y1 In-person Mean (%) Difference | | Y2 Virtual Workshop Mean Difference | | Y2 Online Course Mean Difference | | P-Value* |
|---|---|---|---|---|---|---|---|
| | n | Mean+ (%) | n | mean+ (%) | n | mean+ (%) | |
| Total Score | 3027 | +1.2 (13.6) | 2595 | +0.5 (6.0) | 629 | +0.7 (7.6) | <0.001 |
| Training Level | | | | | | | |
| Pre-service learner | 1755 | +1.3 (14.1) | 1518 | +0.5 (5.1) | 299 | +0.6 (7.0) | <0.001 |
| Post-graduate new provider | 727 | +1.2 (13.4) | 291 | +0.6 (6.5) | 66 | +0.7 (7.4) | <0.001 |
| Post-graduate provider | 540 | +1.1 (12.5) | 384 | +0.7 (7.6) | 158 | +1.0 (11.6) | <0.001 |
| Health Profession | | | | | | | |
| Laboratory | 365 | +1.7 (19.4) | 86 | +0.6 (6.8) | 117 | +1.2 (13.7) | <0.001 |
| Medical | 902 | +1.0 (10.7) | 582 | +0.5 (5.7) | 140 | +0.8 (9.3) | <0.001 |
| Nursing/midwifery | 1145 | +1.4 (15.0) | 949 | +0.5 (5.8) | 107 | +0.4 (4.5) | <0.001 |
| Pharmacy | 312 | +1.1 (12.3) | 373 | +0.5 (5.0) | 95 | +0.6 (6.5) | <0.001 |
| Other | 299 | +1.0 (11.6) | 203 | +0.5 (5.9) | 64 | +0.6 (6.7) | <0.01 |

*Analysis of Variance (ANOVA) test used to compare mean differences in knowledge across the three educational programs. Abbreviations: Y1: Year one; Y2: Year two; post-graduate new provider: Provider participating in training less than 12 months since graduating from health profession training program; Post-graduate provider: Provider participating in training at least 12 months or more since graduating from health profession training program; % diff = Percent difference between the mean pre and the mean post score. Maximum Score = 9.

+Denotes absolute difference in mean scores between post-test and pre-test scores.

learners [p<0.001]; Y1 in-person new graduate learners experienced a mean 13.4% [+1.2 points] increase in knowledge vs. 6.5% [+0.6 points] for Y2 VW learners vs. 7.4% [+0.7 points] for OC learners[p<0.001]; and Y1 in-person post-graduate learners experienced a mean 12.5% [+1.1 points] increase in knowledge vs. 7.6% [0.7 points] for Y2 VW learners vs. 11.6% [+1.0 points] for OC learners [p<0.001]). Among pre-service and post-graduate new provider learners there were no significant differences in knowledge scores noted between VW and OC participants, but there were greater gains in knowledge among post-graduate provider learners who participated in the OC compared to the VW (Tables 2 and S3). Regardless of the health professions cadre, gains in knowledge were greater among Y1 In-person learners compared to learners who participated in the VW or OC formats. Among learners from nursing, pharmacy and other cadres, there was no difference in knowledge gains between participants in the VW and OC formats.

## Confidence assessment

Baseline confidence levels were higher among post-graduate learners with more than 12 months of clinical experience compared to other types of learner. However, across all learner subgroups, self-perceived confidence increased between the pre- and post-course assessments (Table 3, Figs 1 and 2). Overall, the percent increases in confidence scores were significantly lower among learners who participated in the VW or OC than among learners who participated in the Y1 in-person training ((Y1 in-person pre-service learners experienced 18.9% [+13.6 points] mean increase in confidence vs. 15% [+10.8 points] for Y2 VW learners vs. 14.3% [+10.3 points] for OC learners [p<0.001])) (Tables 3 and S4). Gains in confidence were significantly greater among pre-service learners (p<0.001) and post-graduate new provider learners (p<0.001) who participated in Y1 in-person training compared to the VW or OC programs (Fig 1); differences in confidence between the three programs were not significantly different among post-graduate provider learners (p = 0.053). Increases in confidence were

**Table 3. Total learner mean confidence score differences during Y1 in-person Training, Y2 Virtual Workshop, and Y2 Online Course by Total Score, Training Level, Health Profession, and Confidence Type compared using ANOVA.**

| | Y1 In-person Mean (%) Difference | | Y2 Virtual Workshop Mean Difference | | Y2 Online Course Mean Difference | | P-Value* |
|---|---|---|---|---|---|---|---|
| | n | Mean[+] (%) | n | mean[+] (%) | n | mean[+] (%) | |
| Total Score | 3027 | +13.6 (18.9) | 2595 | +10.8 (15.0) | 629 | +10.3 (14.3) | <0.001 |
| Training Level | | | | | | | |
| Pre-service learner | 1755 | +13.6 (18.9) | 1518 | +9.9 (13.8) | 299 | +11.0 (15.3) | <0.001 |
| Post-graduate new provider | 727 | +14.3 (19.8) | 291 | +11.0 (15.2) | 66 | +10.6 (14.8) | <0.001 |
| Post-graduate provider | 540 | +12.8 (17.8) | 384 | +11.7 (16.3) | 158 | +10.4 (14.4) | 0.053 |
| Health Profession | | | | | | | |
| Laboratory | 365 | +10.4 (14.5) | 86 | +11.0 (15.3) | 117 | +11.0 (15.2) | 0.876 |
| Medical | 902 | +15.7 (21.8) | 582 | +14.4 (20.0) | 140 | +13.1 (18.2) | 0.016 |
| Nursing/midwifery | 1145 | +14.2 (19.8) | 949 | +8.5 (11.8) | 107 | +7.5 (10.4) | <0.001 |
| Pharmacy | 312 | +13.4 (18.6) | 373 | +9.6 (13.3) | 95 | +12.0 (16.6) | <0.001 |
| Other | 299 | +9.2 (12.8) | 203 | +8.9 (12.3) | 64 | +9.1 (12.8) | 0.969 |
| Confidence Type | | | | | | | |
| Clinical | 3027 | +11.1 (18.5) | 2595 | +8.8 (14.7) | 629 | +8.3 (13.9) | <0.001 |
| IP | 3027 | +1.4 (16.9) | 2595 | +1.1 (14.3) | 629 | +1.2 (14.6) | <0.001 |
| QI | 3027 | +1.1 (28.7) | 2595 | +0.8 (21.2) | 629 | +0.8 (19.5) | <0.001 |

*Analysis of Variance (ANOVA) test used to compare mean differences in confidence across the three educational programs. Abbreviations: Y1: Year one; Y2: Year two; post-graduate new provider: Provider participating in training less than 12 months since graduating from health profession training program; Post-graduate provider: Provider participating in training at least 12 months or more since graduating from health profession training program; IP: Interprofessional Confidence; QI: Quality Improvement Confidence; % diff = Percent difference between the mean pre and the mean post score.

+Denotes absolute difference in mean scores between post- and pre-scores.

greater among medical, nursing/midwifery, and pharmacy learners participating in Y1 in-person training compared to the VW or OC formats (p = 0.016, p<0.001 and p<0.001 respectively); differences in confidence among laboratorian participants between the three course were not significantly different (p = 0.876) (Fig 2). Regardless of the domain of confidence assessed ((i) clinical confidence, (ii) confidence to engage in Inter-professional practice (IP), and (iii) confidence in quality improvement practice (QI)) there was no significant difference in gains in confidence between participants in the VW or OC formats (S1 Fig).

## Sensitivity analysis

There was a small proportion of participants for whom we had complete demographic data but not evaluative data. We undertook a sensitivity analysis to determine if those participants for whom we did not have complete demographic records responded differently to the pre- and post-surveys from those for whom we did not have complete demographic data. In this additional analysis, S1 Table, we found no differences in mean scores for knowledge or confidence between those for whom we had demographic data and those we did not, regardless of whether they were in the IP, VW or OC group.

## Discussion

In this multi-country capacity-building intervention, which included over 13,000 learners from diverse cadres and clinical settings, in-person and different online educational strategies all led to significant improvements in knowledge and clinical confidence. Moreover, the transition from in-person to online and remote training that occurred because of the COVID-19

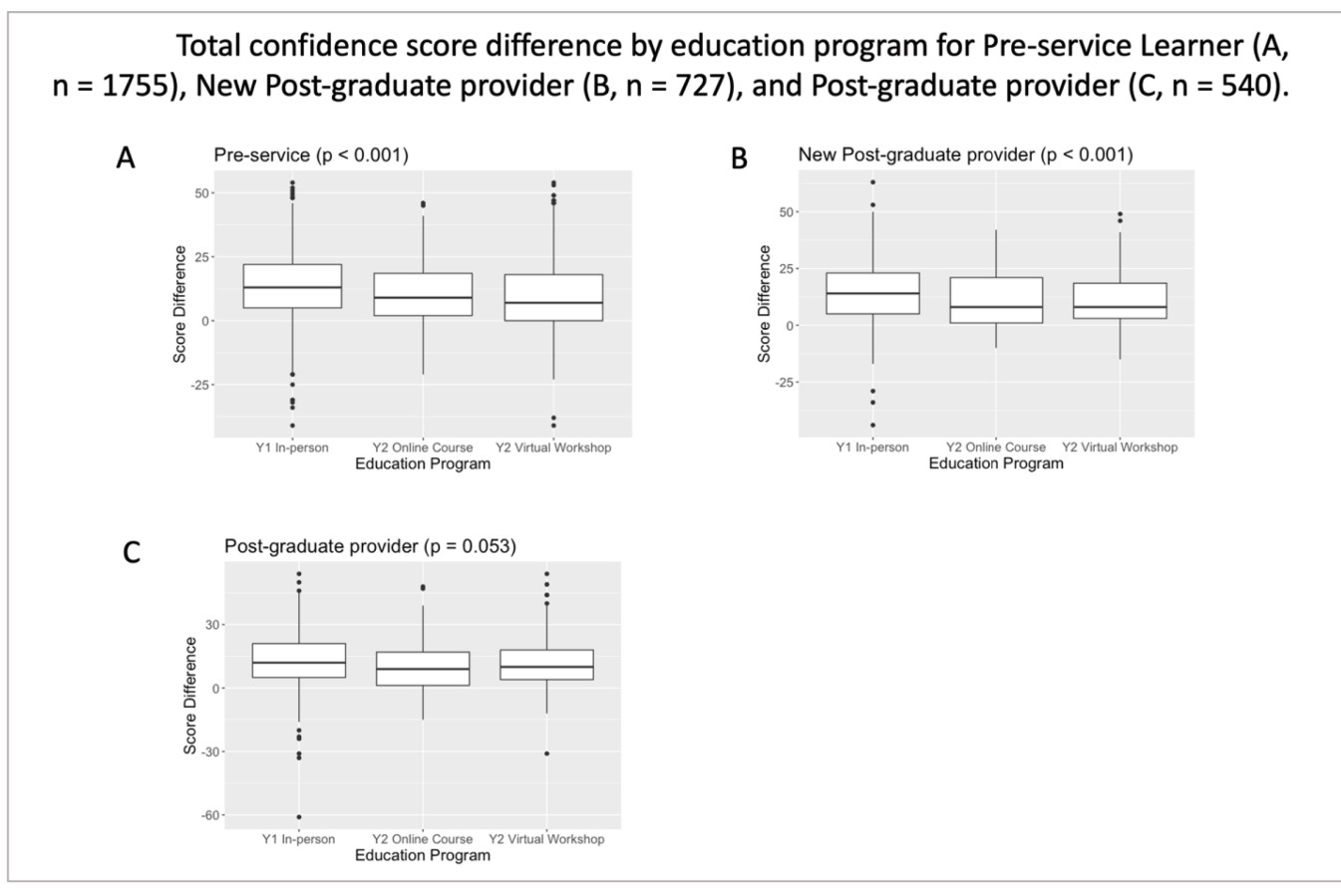

**Fig 1.**

pandemic, did not have a major deleterious impact on learner outcomes, as demonstrated by the gains in both knowledge and confidence across all three educational programs and for all types of learners. Nonetheless, gains in knowledge and clinical confidence were greater among in-service learners in Y1 compared to learners in Y2. These findings have several important programmatic and policy implications, as outlined below.

Firstly, it was notable that the magnitude of gains in technical knowledge and clinical confidence were smaller among learners who participated in the synchronous Virtual Workshop and blended Online Course compared to those in the in-person training. Across all domains of confidence assessed, mean improvements in subjective confidence were greatest among learners who participated in the in-person program, regardless of cadre and career stage. We are cautious drawing conclusions about the implications of these findings on clinical outcomes since this was not assessed in this study. Nonetheless, these findings validate other research demonstrating the importance of online tools as satisfactory for acquiring knowledge, but inferior modalities for acquisition of clinical skills or technical confidence [21,22]. These findings underscore the importance of in-person learning for inter-professional training. They also highlight the potential limitations of on-line training, which may be considerable, if deficiencies in knowledge and confidence lead to adverse clinical outcomes. These results should be weighed against the relative merits of online remote learning modalities. For example, online learning tools are likely to be more affordable in many parts of SSA, especially given that learners can participate even from remote or inaccessible geographic locations.[10] Recent studies in

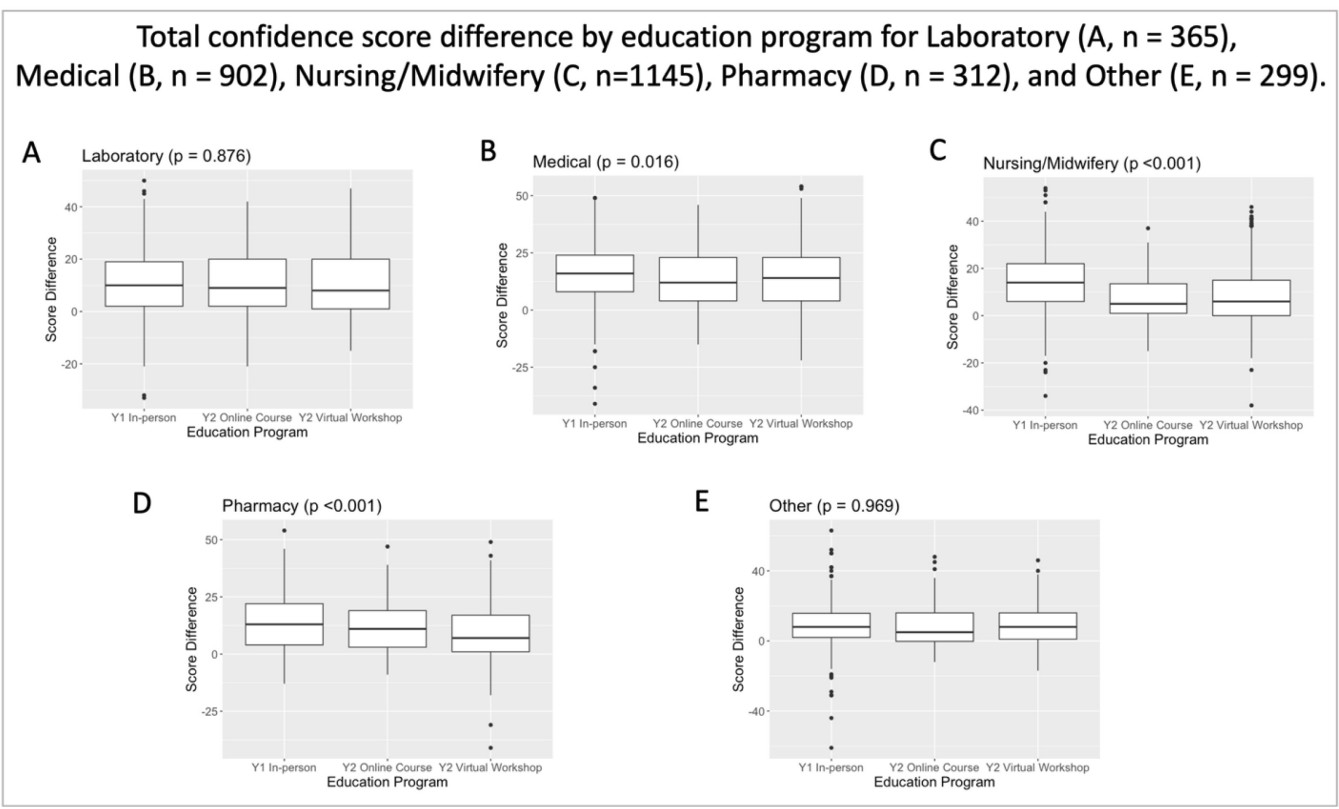

**Fig 2.**

SSA have highlighted substantial opportunity costs associated with in-person training programs for in-service learners that are potentially obviated by distance learning modalities [14,15,23]. These benefits notwithstanding [24], more research is warranted to ensure that online capacity-building interventions are used effectively, and in combination with traditional in-person strategies, to foster clinical confidence and inter-professional collaboration, and determine if they lead to improvements in provider performance and superior clinical outcomes. It is notable that confidence gains were greater for learners who participated in the synchronous Virtual Workshops compared to the blended Online Course format, highlighting that even when in-person workshops may not be feasible or affordable, strategies to facilitate real-time learning using online video-streaming platforms, can be leveraged, even in resource-variable settings in SSA.

Secondly, this analysis underscores how educational technologies can play a critical role in addressing inequities in access to training opportunities in SSA. Despite the profound, disruptive impacts of the COVID-19 pandemic, both the online synchronous and blended programs enabled more diverse and inclusive approaches to inter-professional HIV training for in-service learners. Learners in Y2 were more likely to be female and from more diverse cadres than learners in Y1. These findings are in contrast with other data highlighting how the COVID-19 pandemic revealed and increased pre-existing inequities in professional development opportunities for healthcare professionals in LMICs [2,25,26], Although many health professions training institutions in SSA lack access and capacity to use digital technologies to deliver HIV training [27] our results affirm the critical role that online training interventions can play in advancing professional development opportunities especially for those healthcare professions historically underserved by capacity building interventions [28].

Thirdly, this study highlights the importance of adaptive and resilient educational strategies in response to an emergent infectious disease threat. Although very few of the partner institutions involved in the program had adopted online learning tools before COVID-19, the pandemic accelerated a shift to digital technologies in health professions education even in countries with limited digital infrastructure. Moreover, use of digital platforms allowed learners in diverse settings to acquire new knowledge rapidly and effectively. While further research is critical to better understand how to optimally leverage digital tools in combination with more immersive in-person teaching formats, our study offers an African-specific blueprint for how to rapidly upskill the healthcare workforce, even in the setting of a major public health emergency.

The study has several limitations. Firstly, data is from an HIV-specific training program, and evaluative data limited to learners who completed the registration survey and assessment questions. Further research is warranted to determine if lessons learned from the analysis are applicable to other disease-specific capacity-building interventions. The three different approaches employed different pedagogical techniques (for example both in-person and virtual workshops included small group breakouts) that might have impacted learner outcomes as much as the medium of delivery (online vs. in-person). More health professions education research to understand how best to employ digital tools in Africa is a priority that has been highlighted by numerous stakeholders [16,29–31]. Additionally, our assessment was limited to knowledge and confidence, as data regarding clinical outcomes was not available. Since different health professional training institutions participated in the program in Y1 and Y2 (in part due to digital literacy and internet infrastructure), we were also unable to compare learner outcomes between Y1 and Y2 within individual institutions or country. Nonetheless, we assert that the aggregate findings have broad applicability to diverse Africa settings. Finally, institutions in Y2 were different from Y1 since participation was limited by digital literacy and Internet infrastructure. In Y2, we only included information from learners that were able to participate in the digital modalities highlighted here; the study does not address barriers faced by learners unable to access online training programs in SSA. These limitations highlight substantial digital inequities in health professions training institutions between and within countries in SSA [5,25], and underscore the ongoing need to support the development of local technological infrastructure for online education across SSA.

## Supporting information

**S1 Fig. Confidence score difference by education program for Clinical confidence (A, n = 3027), Interprofessional Confidence (B, n = 3027), and Confidence to incorporate Quality Improvement (Ql) into clinical practice (C, n = 3027).**
(TIF)

**S1 Table. Assessment questions for each module.**
(DOCX)

**S2 Table. The maximum score for each assessment category.**
(DOCX)

**S3 Table. Total learner baseline mean knowledge score differences for each educational program (Y1 In-person, Y2 Virtual Workshop, and Y2 Online Course) by Total Score, Training Level, and Health Profession compared using Wilcoxon Signed Rank Test.**
(DOCX)

**S4 Table. Total learner baseline mean confidence score differences for each educational program (Y1 In-person, Y2 Virtual Workshop, and Y2 Online Course) by Total Score and**

Confidence Type using Wilcoxon Signed Rank Test.
(DOCX)

## Author Contributions

**Conceptualization:** E. Kiguli-Malwadde, M. Forster, J. Celentano, S. Martin, R. Mubuuke, D. Sears, D. von Zinkernagel, M. J. A. Reid, F. Suleman.

**Data curation:** M. Forster.

**Formal analysis:** M. Forster, R. Mubuuke, D. Sears, M. J. A. Reid, F. Suleman.

**Funding acquisition:** J. Celentano.

**Investigation:** E. Chilembe, I. D. Couper, E. T. Dassah, M. R. De Villiers, O. Gachuno, C. Haruzivishe, J. Khanyola, K. Motlhatlhedi, K. A. Mteta, P. Moabi, A. Rodrigues, F. Semitala.

**Methodology:** E. Chilembe, I. D. Couper, E. T. Dassah, M. R. De Villiers, O. Gachuno, C. Haruzivishe, J. Khanyola, K. Motlhatlhedi, K. A. Mteta, P. Moabi, A. Rodrigues, F. Semitala.

**Project administration:** D. von Zinkernagel, M. J. A. Reid.

**Resources:** M. J. A. Reid.

**Supervision:** E. Kiguli-Malwadde, F. Suleman.

**Visualization:** M. Forster.

**Writing – original draft:** E. Kiguli-Malwadde, M. Forster, A. Eliaz, S. Martin, D. von Zinkernagel, M. J. A. Reid, F. Suleman.

**Writing – review & editing:** E. Kiguli-Malwadde, M. Forster, I. D. Couper, E. T. Dassah, M. R. De Villiers, O. Gachuno, C. Haruzivishe, J. Khanyola, S. Martin, K. Motlhatlhedi, R. Mubuuke, K. A. Mteta, P. Moabi, A. Rodrigues, D. Sears, F. Semitala, M. J. A. Reid, F. Suleman.

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
