## [Decision Letter · Decision Letter 0]

27 Mar 2023

PGPH-D-23-00199

Comparing in-person, blended and virtual training interventions; a real-world evaluation of HIV capacity building programs in 16 countries in sub-Saharan Africa

Dear Dr. Reid,

Thank you for submitting your manuscript to PLOS Global Public Health. After careful consideration, we feel that it has merit but does not fully meet PLOS Global Public Health’s publication criteria as it currently stands. Therefore, we invite you to submit a revised version of the manuscript that addresses the points raised during the review process.

Note the important points raised on review, especially:

- inclusion of the range of recent literature on this topic

- detailed consideration of how differences between participants affects results in addition to differences in training, not currently addressed in the analysis shown

- clarity in interpreting differences in pre/post scores, which are not evidence of improved clinical effectiveness, and equally caution in generalizing from this study to online training in general

We look forward to receiving your revised manuscript.

Kind regards,

Hannah Hogan Leslie, PhD

Academic Editor

Journal Requirements:

1. Please provide additional details regarding participant consent. In the ethics statement in the Methods and online submission information, please ensure that you have specified (1) whether consent was informed and (2) how the verbal consent was documented and witnessed.

2. Please also include a complete copy of PLOS’ questionnaire on inclusivity in global research in your revised manuscript. Our policy for research in this area aims to improve transparency in the reporting of research performed outside of researchers’ own country or community. The policy applies to researchers who have travelled to a different country to conduct research, research with Indigenous populations or their lands, and research on cultural artefacts. The questionnaire can also be requested at the journal’s discretion for any other submissions, even if these conditions are not met. Please find more information on the policy and a link to download a blank copy of the questionnaire here: https://journals.plos.org/globalpublichealth/s/best-practices-in-research-reporting. Please upload a completed version of your questionnaire as Supporting Information when you resubmit your manuscript.

3. Please send a completed 'Competing Interests' statement, including any COIs declared by your co-authors. If you have no competing interests to declare, please state "The authors have declared that no competing interests exist". Otherwise please declare all competing interests beginning with the statement "I have read the journal's policy and the authors of this manuscript have the following competing interests:"

Additional Editor Comments (if provided):

1) Please clarify the training methods. For instance, the abstracts describes the workshop as asynchronous, while the methods section says synchronous. Provide details on the time spent in each method, the inclusion of breakout groups, and the size of such groups to provide greater depth and clarity on the pedagogical similarities and differences beyond delivery mode.

2) Please include adequate detail in the methods on outcomes and analytic approach; are the pre- and post-test quizzes the same exact items? Is the time between pre and post tests equivalent across the training methods? Include the baseline scores in reporting results and clarify linear (absolute) or relative % differences in reporting

3) Note there is substantial attrition or non-response in the online participants - this should be described more clearly, with available information on demographics reported as supplement if possible, and consideration on the difference between those enrolling and those completing post-test assessments added to the limitations

4) The analysis can be more nuanced to address the differences in learners, institutions, and countries. Consider multivariable analysis as suggested by one reviewer or a decomposition analysis to identify whether the differences in mean score difference by learner group are attributable to training mode vs. characteristics of the participants. A subanalysis limiting to the countries with data on each mode would also help to control for potential differences in trainees and trainers across countries. These analyses would add substantially to the inference.

Reviewers' comments:

Reviewer's Responses to Questions

**Comments to the Author**

1. Does this manuscript meet PLOS Global Public Health’s publication criteria? Is the manuscript technically sound, and do the data support the conclusions? The manuscript must describe methodologically and ethically rigorous research with conclusions that are appropriately drawn based on the data presented.

Reviewer #1: Partly

Reviewer #2: Yes

2. Has the statistical analysis been performed appropriately and rigorously?

Reviewer #1: No

Reviewer #2: Yes

3. Have the authors made all data underlying the findings in their manuscript fully available (please refer to the Data Availability Statement at the start of the manuscript PDF file)?

Reviewer #1: Yes

Reviewer #2: Yes

4. Is the manuscript presented in an intelligible fashion and written in standard English?

Reviewer #1: Yes

Reviewer #2: Yes

5. Review Comments to the Author

Reviewer #1: The authors have leveraged a natural experiment due to COVID-19 which required the rapid transition from in person to 2 models of virtual training for HIV providers both in-service and pre-service. The authors make good use of the available data and many of the limitations are discussed.

The background discusses the need to switch as well as the limited evaluation to date . There is now a wealth of literature, although many of these may have been after submission. This should be updated.

However there were a number of areas where additional clarity ss needed in sections.

Methods.

1. The details in analysis at the bottom of the tables belongs in methods.

2. The decision of which option was made by the institution and a discussion of differences in those settings as well as the providers they train would have been important beyond the individual level differences.

3. There were multiple comparison between demographics, but no discussion of multiple tests.

Results

1. When describing the results, % is needed as well as numbers, and ideally some description if available of the differences between people who responded and who did not

2. The baseline scores on knowledge and confidence across the 3 groups would also help understand additional differences

3. I assume that the differences are absolute but this should be defined

4. Consideration of a MVA would have helped understand if the differences were due to the mode versus the attendees.

5. For the confidence, was there any differences in attendance at the relative sessions (or were those 4 the ones related to the confidence)

Discussion

1. The first sentence that were “similarly effective” does not reflect the main findings, although it is clear that the differences may not have been proportionally difference (relative versus absolute change)

2. The first sentence of paragraph 2 seems incomplete

3. The virtual workshop has breakouts, while the online classroom and in-person did not. So comparison of the 2 virtual approaches needs to address differences in difference forms of pedagogy

4. References to other work for transition to online training during COVID as well as the established literature on online training is needed.

Reviewer #2: Thank you for inviting me to review this paper. The manuscript is well written and succinct. During the COVID pandemic there was a huge need to continue education of the health workforce in all disciplines and many contributed to these efforts. What remains to be done is assessment of the efficacy of these virtual interventions and this paper is an excellent first step. My clinical area of expertise is maternal health and I have been an instructional designer for the past 10 years. My comments on the paper are through this lens. I have a few copy edit suggestions and the rest are more substantive.

Overall:

- I think we want to take great care in calling training effective when we are not truly measuring the goal of training - change in provider performance to improve health outcomes. I think we must be very careful not to give the impression that virtual learning is the panacea we've all been waiting for. This paper does not say that but I believe it could be a bit stronger in saying what research is needed and what we cannot say with knowledge and confidence assessments. You'll see below specific places you could consider strengthening that language.

- Interesting but not surprising finding that online = more equity in some respects. However where bandwidth is not good, virtual learning is inaccessible so a different type of inequity.

PDF Pages:

8 - I would list the countries after "participated"

8 - hitherto means before - I believe you might use hereafter?

9 - "To compare the three learning approaches, we evaluated data from four training four modules that

utilized" Did you mean to use four twice?

12 - Greater improvement in knowledge gains in-person compared to VW/OC seems to show the value of in person. OR is it really the difference in how many hours for in person vs online meaning engagement with the learning? OR changes in materials? You describe the tests as being the same but a bit of a description of # of hours of each recognizing that you would have to estimate as asynchronous learning varies. For in person I believe you said 2 days so is this 16 hours? Similarly, how were the materials changed/adapted for the VW/OC. If they were shortened, this could explain some of the differences you saw.

13 - Is the change in confidence less in post grads likely b/c post grad start more confident? If so, would be good to mention and discuss

14 - Interesting that the difference for clinical confidence is huge. Might be worth discussing? Were the training materials more geared to the clinical components?

Probably the biggest area that I'd like to see fleshed out a bit more concerns changes in knowledge and confidence being equated with efficacy. The literature is replete with examples showing improvements in knowledge and even skills after training as compared to before but then NO change in provider performance or outcomes occurs.

14 - "in-person and different online educational strategies were demonstrated to be similarly effective means of training learners in inter-professional HIV care across SSA." Effective by what metric? So the two you measured yes but I think we have to state early and often that this does not automatically equate to efficacy for the real goal which is improving care for patients with HIV.

14-15 - "Moreover, the transition from in-person to online and remote training that occurred because of the COVID-19 pandemic, did not have a major deleterious impact on learner outcomes, as demonstrated by the gains

in both knowledge and confidence across all three educational programs and for all types of learners." But the gains were not as good for remote on knowledge and we don't know about the learner outcome of change in performance.

15 "For example, online learning tools are likely to be more affordable in many parts of SSA" Yes, but I think we need to be careful stating this until we have knowledge that the effect on patient care and outcomes is similar. Something may be affordable but it it's not effective it will never be cost effective.

15 "These benefits notwithstanding,(17) more research is warranted to ensure that online capacity building interventions are used effectively, and in combination with traditional inperson strategies, to foster clinical confidence and inter-professional collaboration." I agree 100% but please add something about competence and/or change in provider performance.

16 - "Although many health professions training institutions in SSA lack access and capacity to use digital technologies to deliver HIV training(20) our results affirm the critical role that online training interventions can play in advancing professional development opportunities especially for those healthcare professions historically underserved by capacity building interventions.(21)" Yes, but we need to ensure that the end goal - improving patient care through change in provider behaviors - is achieved. So many donors and even ministries now are saying that training can all be done virtually and we know that this is not an accurate statement. I would never train someone to conduct a birth, manage a hemorrhage, use a vacuum for an assisted delivery exclusively online. However how far can we go to decrease the face to face time through blended learning? Blended approaches are indeed needed and can be beneficial. What is is the right balance of in person vs virtual? While HIV care is likely much less skill driven as it is adherence to a protocol or treatment algorithm, something made in person learning more effective at knowledge acquisition.

6. PLOS authors have the option to publish the peer review history of their article (what does this mean?). If published, this will include your full peer review and any attached files.

**Do you want your identity to be public for this peer review?** For information about this choice, including consent withdrawal, please see our Privacy Policy.

Reviewer #1: No

Reviewer #2: **Yes: **Cherrie L Evans DrPH, CNM

---

## [Decision Letter · Decision Letter 1]

8 Jun 2023

Comparing in-person, blended and virtual training interventions; a real-world evaluation of HIV capacity building programs in 16 countries in sub-Saharan Africa

PGPH-D-23-00199R1

Dear Dr Reid,

We are pleased to inform you that your manuscript 'Comparing in-person, blended and virtual training interventions; a real-world evaluation of HIV capacity building programs in 16 countries in sub-Saharan Africa' has been provisionally accepted for publication in PLOS Global Public Health.

Best regards,

Hannah Hogan Leslie, PhD

Academic Editor

Reviewer Comments (if any, and for reference):

Reviewer's Responses to Questions

**Comments to the Author**

1. If the authors have adequately addressed your comments raised in a previous round of review and you feel that this manuscript is now acceptable for publication, you may indicate that here to bypass the “Comments to the Author” section, enter your conflict of interest statement in the “Confidential to Editor” section, and submit your "Accept" recommendation.

Reviewer #2: All comments have been addressed

2. Does this manuscript meet PLOS Global Public Health’s publication criteria? Is the manuscript technically sound, and do the data support the conclusions? The manuscript must describe methodologically and ethically rigorous research with conclusions that are appropriately drawn based on the data presented.

Reviewer #2: Yes

3. Has the statistical analysis been performed appropriately and rigorously?

Reviewer #2: N/A

4. Have the authors made all data underlying the findings in their manuscript fully available (please refer to the Data Availability Statement at the start of the manuscript PDF file)?

Reviewer #2: Yes

5. Is the manuscript presented in an intelligible fashion and written in standard English?

Reviewer #2: Yes

6. Review Comments to the Author

Reviewer #2: (No Response)

7. PLOS authors have the option to publish the peer review history of their article (what does this mean?). If published, this will include your full peer review and any attached files.

**Do you want your identity to be public for this peer review?** For information about this choice, including consent withdrawal, please see our Privacy Policy.

Reviewer #2: **Yes: **Cherrie Lynn Evans, DrPH, CNM
